# Cefiderocol Antimicrobial Susceptibility Testing by Disk Diffusion: Influence of Agar Media and Inhibition Zone Morphology in *K. pneumoniae* Metallo-β-lactamase

**DOI:** 10.3390/antibiotics14050527

**Published:** 2025-05-21

**Authors:** Maciej Saar, Anna Wawrzyk, Dorota Pastuszak-Lewandoska, Filip Bielec

**Affiliations:** 1Department of Microbiology and Laboratory Medical Immunology, Medical University of Lodz, 90-151 Lodz, Poland; 2Department of Basic Biomedical Science, Faculty of Pharmaceutical Sciences in Sosnowiec, Medical University of Silesia, 41-205 Sosnowiec, Poland

**Keywords:** antimicrobial susceptibility testing, area of technical uncertainty, cefiderocol, colony interpretation, disk diffusion, inhibition zone morphology, *Klebsiella pneumoniae*, metallo-β-lactamase

## Abstract

Accurate antimicrobial susceptibility testing (AST) of cefiderocol remains a diagnostic challenge, especially in infections caused by metallo-β-lactamase (MBL)-producing *Klebsiella pneumoniae*. While disk diffusion offers a cost-effective alternative to broth microdilution, it is highly sensitive to factors such as media composition and the presence of atypical colony morphology. The objective of this study was to evaluate how different agar media and interpretations of isolated colonies affect the performance and reliability of cefiderocol AST by disk diffusion. A total of 50 clinical *K. pneumoniae* MBL isolates were tested using disk diffusion on Columbia with blood, MacConkey, and chromogenic agars from three manufacturers. Inhibition zones were compared with MICs from broth microdilution. Statistical analyses included paired *t*-tests and Spearman correlation to assess media effects and zone morphology impact. Variability in inhibition zone diameters was observed between media, notably with chromogenic agar. The most consistent results were obtained using Graso Biotech and Thermo Fisher Columbia with blood agar. Isolated colonies were observed in over half the samples and, depending on how they were interpreted, led to major changes in classification accuracy. Up to 64% of results fell into the EUCAST area of technical uncertainty (ATU), and categorical agreement varied across media and interpretive criteria. Disk diffusion for cefiderocol may be used in resource-limited settings but only if rigorously standardized using validated media, consistent zone reading, and ATU-aware interpretive strategies. In borderline cases or when morphological anomalies are present, broth microdilution should be considered the sole reliable method. Clinical microbiologists are advised to exercise caution with ambiguous results and seek expert or confirmatory testing when needed.

## 1. Introduction

Multidrug resistance (MDR) among Gram-negative bacteria is one of the greatest challenges in modern medicine. In its 2024 report [1], the World Health Organization (WHO) once again classified *Klebsiella pneumoniae* as a high-priority pathogen due to its ability to develop resistance to multiple classes of antibiotics, including carbapenems, which were previously considered the last line of defense against MDR infections. The emergence of carbapenem-resistant *K. pneumoniae* is largely attributed to the production of carbapenemases, a group of β-lactamases that inactivate carbapenems [2].

Carbapenemases can be classified into three major groups based on the Ambler classification [3]:Class A carbapenemases (e.g., KPC)–these enzymes hydrolyze carbapenems but can be inhibited by certain β-lactamase inhibitors;Class B metallo-β-lactamases (MBLs) (e.g., NDM, VIM, IMP)–these enzymes require zinc ions for activity and are resistant to available for treatment β-lactamase inhibitors;Class C carbapenemases (e.g., OXA-48-like)–these are particularly problematic due to their ability to hydrolyze a broad range of β-lactams [4].

Among these, MBLs warrant special attention due to their ability to confer high levels of resistance to nearly all β-lactams while also being unaffected by traditional β-lactamase inhibitors [2]. The global dissemination of MBL-producing *K. pneumoniae* poses a significant threat to public health, limiting therapeutic options and increasing mortality rates among infected patients [2,4].

A promising solution against carbapenem-resistant *K. pneumoniae* is cefiderocol—a novel cephalosporin–siderophore conjugate antibiotic specifically designed to target MDR Gram-negative bacteria. Unlike traditional cephalosporins, cefiderocol mimics siderophores—molecules used by bacteria to acquire iron from their environment. By binding to ferric iron, cefiderocol exploits bacterial iron transport systems to facilitate its active uptake across the outer membrane, allowing it to evade many resistance mechanisms, including porin mutations and efflux pumps [5,6]. Cefiderocol has demonstrated potent in vitro activity against a broad range of MDR Gram-negative pathogens, including carbapenemase-producing Enterobacterales and non-fermenters (e.g., *Pseudomonas aeruginosa*, *Acinetobacter baumannii*). Given its novel mechanism and broad activity, cefiderocol represents a significant opportunity in the fight against MDR infections, particularly those caused by MBL-producing *K. pneumoniae*, for which few treatment options remain [6,7,8,9].

### 1.1. Challenges in Cefiderocol Susceptibility Testing

Despite its promising antimicrobial activity, the routine use of cefiderocol in clinical settings is hindered by significant challenges, including antimicrobial susceptibility testing (AST). Several biological, technical, economic, and organizational factors influence its reliability.

#### 1.1.1. Biological Factors Affecting Susceptibility Testing

Cefiderocol’s activity is influenced by the availability of iron in the testing medium. Since it relies on iron transport for entry into bacterial cells, iron-depleted conditions are necessary for accurate susceptibility testing. Standard laboratory media, such as Mueller-Hinton agar (MHA), may contain variable iron concentrations, potentially leading to inconsistent results [10]. European Committee on Antimicrobial Susceptibility Testing (EUCAST) and Clinical and Laboratory Standards Institute (CLSI) recommend using iron-depleted media to ensure the reproducibility of cefiderocol susceptibility testing [11,12,13].

Another biological factor particularly relevant for cefiderocol is the inoculum effect—a phenomenon where higher bacterial loads lead to increased minimal inhibitory concentration (MIC) values. Studies suggest that elevated bacterial densities can impair drug efficacy by overwhelming iron transport systems or enhancing bacterial defense mechanisms. This necessitates strict adherence to standardized bacterial inoculum sizes in AST procedures [14].

#### 1.1.2. Technical Challenges in Standardizing Cefiderocol AST

Unlike other β-lactams, cefiderocol does not have a universally agreed-upon AST method. The three primary methods used for testing cefiderocol susceptibility include the following:Broth microdilution—the reference method recommended by EUCAST and CLSI, but labor-intensive and costly.Gradient strip diffusion—provides an MIC value but may be affected by iron availability.Disk diffusion test—the most accessible method but subject to an area of technical uncertainty (ATU), meaning that certain zone diameter results are difficult to interpret reliably [15,16].

Another technical challenge is the appropriate interpretive criteria, as different organizations have established varying cefiderocol breakpoints, leading to potential inconsistencies. EUCAST defined separate breakpoints for Enterobacterales and non-fermenters, incorporating ATU ranges [11], while CLSI established different breakpoints and did not account for ATU [12]. A summary of current cefiderocol breakpoints for Enterobacterales is presented in Table 1.

The presence of ATU in the EUCAST criteria suggests that some susceptibility results may be unreliable and require additional confirmatory testing or reporting as resistant [16].

#### 1.1.3. Economic Barriers to Cefiderocol AST

Many hospital laboratories operate under limited budgets, making the implementation of advanced AST methods challenging. The high cost of broth microdilution poses a significant financial barrier. Gradient strip tests, while providing an MIC value, are not recommended for routine diagnostics due to their limited accuracy in cefiderocol susceptibility testing [15]. EUCAST and CLSI do not endorse gradient strip methods for cefiderocol AST, and their use should be avoided in routine diagnostics [11,12]. To facilitate the routine use of cefiderocol in clinical settings, it is essential to develop low-cost, accessible AST methods that maintain accuracy and reproducibility. Disk diffusion, despite its limitations, remains a viable option for resource-limited settings if appropriate interpretive criteria are established [16].

#### 1.1.4. Organizational Factors Affecting AST in Clinical Laboratory

To optimize bacterial identification and AST workflow in clinical microbiology laboratories, various culture media are used. While blood agar remains a standard choice, many laboratories also utilize selective-differential and chromogenic media to accelerate and simplify diagnostics [17]. Selective media contain agents that inhibit the growth of certain bacteria while allowing the target organisms to proliferate, facilitating pathogen isolation [18]. Chromogenic media, on the other hand, incorporate substrates that produce distinct color changes based on bacterial enzymatic activity, aiding in the rapid differentiation of species [17,18]. However, the components and additives in these specialized media can alter bacterial growth dynamics and antimicrobial activity, and, in consequence, influence AST results, potentially affecting the interpretation of susceptibility patterns [19].

### 1.2. Study Objective

This study aimed to evaluate the impact of different microbiological media on cefiderocol susceptibility testing using disk diffusion methodology. The research compared AST results for clinical *K. pneumoniae* MBL isolates previously cultured on various media, including Columbia Agar with 5% Blood (CAB), MacConkey Agar (MCA), and Chromogenic Agar (CHA) from three different manufacturers.

The composition of culture media plays a crucial role in antimicrobial susceptibility testing. CAB, MCA, and CHA differ significantly in terms of, e.g., peptone content, buffering capacity, and the presence of additives like bile salts or chromogenic substrates. These biochemical properties may alter bacterial growth dynamics, zone morphology, and antibiotic diffusion. For instance, CHA can introduce visual artifacts or interact with iron-dependent mechanisms, while selective agents in MCA may stress bacterial metabolism and affect susceptibility patterns. The selected media in this study were also chosen for their routine use in clinical diagnostics.

By assessing the influence of culture conditions on inhibition zone diameters, this study has sought to provide insights that contribute to the standardization of cefiderocol AST methods and improve diagnostic reliability.

## 2. Results

### 2.1. Differences Between Agar Media

The inhibition zone diameters for cefiderocol disk diffusion testing in 50 clinical *K. pneumoniae* MBL isolates showed minimal but measurable variation across different culture media and manufacturers. Paired *t*-tests revealed that most comparisons between CAB, MCA, and CHA did not yield statistically significant differences. However, select comparisons indicated potential media-dependent effects. Significant differences were observed only in four specific pairings—see Table 2.

Spearman’s correlation coefficients between MIC values (from broth microdilution) and inhibition zone diameters were generally negative, indicating the expected inverse relationship: as inhibition zones decreased, MICs increased. The strongest correlations were observed when ignoring single colonies within the inhibition zone and when considering only colonies above 10—see Table 3.

### 2.2. Antimicrobial Susceptibility Interpretation

Using broth microdilution as the reference method, according to current EUCAST and CLSI criteria [10,11], the distribution of cefiderocol susceptibility among the studied 50 *K. pneumoniae* MBL isolates showed a higher rate of resistant classifications by EUCAST criteria compared to CLSI. As shown in Figure 1, EUCAST criteria resulted in a higher proportion of isolates being categorized as resistant (12%) compared to CLSI, with only 4% isolates categorized as intermediate.

The accuracy of cefiderocol disk diffusion susceptibility testing was evaluated in comparison to the broth microdilution reference method, using both EUCAST and CLSI standards. As illustrated in Figure 2, categorical agreement between disk diffusion and broth microdilution varied slightly across media–manufacturer combinations and different approaches to single colonies within the inhibition zone. When ATU was ignored, better accuracy was achieved by ignoring single colonies rather than considering only those exceeding 10 within the inhibition zone. On the other hand, when ATU was included, the number of errors and ATU-classified outcomes did not differ significantly depending on the medium-manufacturer combination or the colony interpretation approach. Based on CLSI criteria, no notable differences were observed across the comparison groups. To ensure statistical robustness and minimize type I errors in multiple comparison settings, we focused our categorical agreement analysis on the six media-manufacturer combinations that showed the strongest and statistically significant correlations between MIC and inhibition zone diameter (see Table 3). These were selected a priori to avoid overinterpretation of weak or non-significant relationships and to reduce false-positive susceptibility classification.

## 3. Discussion

The findings of this study highlight several diagnostic and methodological challenges in cefiderocol susceptibility testing, particularly in relation to MBL-producing *K. pneumoniae*. While cefiderocol remains one of the few therapeutic options for combating carbapenem-resistant Enterobacterales, laboratory AST strategies must be carefully optimized to provide accurate and clinically relevant results. Our analysis focused on several factors influencing AST results, including agar medium composition and the handling of isolated colonies within inhibition zones.

A noticeable divergence in susceptibility classification was observed depending on whether EUCAST or CLSI interpretive criteria were applied. Based on broth microdilution data, EUCAST categorized 12% of isolates as resistant, while CLSI classified all but 4% as susceptible, with the remaining falling into the “intermediate” category (see Figure 1). This discrepancy reflects inherent differences in breakpoints and interpretive philosophies: EUCAST tends to adopt more conservative thresholds, particularly for novel agents and MBL-producing organisms. A similar pattern was previously reported by Isler et al., who found that use of EUCAST breakpoints resulted in resistance classification in 81% of NDM-producing *K. pneumoniae* bloodstream isolates, compared to only 12% using the CLSI criteria [20]. Such inconsistencies have direct clinical implications—laboratories using CLSI may inadvertently underreport resistance, while those using EUCAST may overestimate it, particularly in borderline or ATU-range cases.

One of the key insights from our dataset concerns the effect of agar media on inhibition zone diameters in disk diffusion testing. Although most comparisons between media did not yield statistically significant differences, certain combinations—particularly those involving CHA and MCA—led to measurable shifts in zone sizes that were sufficient to alter susceptibility categorization in borderline cases (see Table 2). These findings are consistent with prior evaluations by others, who demonstrated that commercially available MHAs can produce markedly different zone diameters for antimicrobials, including cefiderocol [15,21,22]. This variability is likely attributable to biochemical properties of the media, including iron content, pH, and solidification agents, all of which may modulate the diffusion and biological activity of cefiderocol, a siderophore antibiotic dependent on iron transport systems for uptake [6]. In our study, Graso Biotech and Thermo Fisher media exhibited the most consistent performance, whereas chromogenic formulations introduced the greatest variability, possibly due to interactions between chromophores, bacterial metabolism, and antibiotic movement through the agar. These observations underscore the critical importance of in-house validation of test media before routine use, particularly when testing compounds with physicochemically sensitive mechanisms. The EUCAST Disk Diffusion Manual explicitly recommends that laboratories verify the quality and performance characteristics of each media batch locally, to ensure test reliability—especially for agents whose susceptibility profiles are known to be influenced by environmental parameters [15,21,22,23].

Our comparison of various strategies for assessing isolated colonies within the inhibition zone—ignoring, counting all, or stratifying by quantity—revealed that excluding small numbers of isolated colonies (≤10 per zone) resulted in the highest correlation with broth microdilution MIC values (see Table 3). This finding is consistent with the general observation that the growth of isolated colonies can complicate zone interpretation but may not reflect stable resistance mechanisms [24].

Diagnostic inconsistencies in our results arose most notably in isolates where minor morphological features, such as isolated colonies within inhibition zones or medium-induced diameter, shifts led to misclassification of cefiderocol susceptibility. The rate of classification errors observed in our study was sufficient to question the diagnostic reliability of disk diffusion for cefiderocol in its current form, especially without confirmatory methods (see Figure 2). For instance, Bovo et al. reported categorical agreement of 92% between disk diffusion and broth microdilution, with major errors (MEs) as high as 16.7% and very major errors (VMEs) of 17.9% when comparing strip tests to broth microdilution, especially in isolates with MIC values close to established breakpoints [25]. Similarly, Morris et al. noted discrepancies between disk diffusion and broth microdilution results, emphasizing the occurrence of inner zone growth and variability in zone interpretation, which complicated accurate susceptibility classification [24]. Our findings are consistent with prior observations, which demonstrated that deviations from standardized test conditions—including differences in antimicrobial source formulation and excipients—can substantially alter both MIC and zone diameter outcomes, leading to potential misclassification [19]. These observations resonate with the findings of Bhalodi et al., who reported significant variability in broth microdilution testing across laboratories, with agreement rates dropping below 70% in certain cases [26]. These examples highlight the need for interpretive frameworks that go beyond rigid categorical thresholds and account for procedural and morphological variability in test results.

Although breakpoint-based categorization was not formally applied in our study, the concept of the ATU remains highly relevant to the interpretation of borderline inhibition zone diameters. In our dataset, 30 to 32 isolates per medium (60–64%) fell within the ATU range defined by current EUCAST guidance (21–23 mm) when isolated colonies within inhibition zones were ignored, and a similarly high proportion—28 to 31 isolates (56–62%)—fell into the same range when only single colonies above 10 per zone were considered (see Figure 2). While we did not assign formal ATU status, the interpretive ambiguity in these borderline cases was evident, especially when overlaid with inter-medium variability and variable growth patterns. These findings underscore the risk of misclassification and diagnostic uncertainty in laboratories relying solely on disk diffusion without access to broth microdilution confirmation. This interpretation aligns with the results of Bianco et al., who reported that 20.5% of Enterobacterales isolates falling into the ATU range by disk diffusion were actually resistant by both the ComASP commercial panel and reference broth microdilution, emphasizing the diagnostic pitfalls when relying solely on inhibition zone diameters [27]. As noted by Brauncajs et al., results within ATU range are disproportionately more frequent when testing MBL-producing *K. pneumoniae* using disk diffusion, often leading to underreporting of true susceptibility [8]. Similar conclusions were drawn by Stracquadanio et al., who observed that ATU values were associated with high rates of false resistance classification [10], and by Castillo-Polo et al., who found that over 70% of tested *K. pneumoniae* isolates fell into the ATU zone with notable inconsistency in interpretation [28]. Baltas et al. reported that nearly 70% of NDM-producing isolates in their UK cohort were resistant to cefiderocol, with an additional 18% falling within the ATU zone—despite no previous cefiderocol exposure. Such findings suggest that MBL-mediated resistance mechanisms may compromise interpretive clarity in disk-based methods [29]. Taken together, these findings—along with our observations—support the view that borderline zone diameters should not be treated as definitive results but instead prompt confirmatory testing, expert review, or provisional reporting with appropriate annotation.

Our data further support the argument for integrated AST interpretation, one that combines phenotypic patterns with procedural factors such as media type, colony morphology, and laboratory reproducibility. Such a multifactorial approach is particularly relevant for cefiderocol, a compound whose susceptibility profile is influenced by complex physicochemical dynamics and bacterial iron acquisition systems. The variability introduced by different testing conditions—e.g., media formulations, antibiotic source materials, or visual interpretation of inhibition zones—demonstrates that a rigid binary model of susceptible versus resistant may obscure clinically relevant nuances. As our study shows, even subtle procedural differences (such as counting or ignoring isolated colonies) can significantly impact zone diameter readings and final classification. In organisms with dynamic resistance expression, such as MBL-producing *K. pneumoniae*, reliance on a single interpretive criterion may lead to misclassification. Therefore, laboratories should not only report raw zone measurements but also contextualize them with information about media type, presence of atypical colony growth, and proximity to ATU cutoffs. As noted by Ersoy et al. and Heithoff et al., expanding AST interpretation with risk assessments or confirmatory testing strategies could mitigate diagnostic uncertainty. The goal is not to complicate interpretation, but to ensure that laboratory data reflect underlying biological variability and testing limitations. This is particularly important for new or last-resort antibiotics, the misinterpretation of which may lead to both therapeutic failure and unintended resistance selection [30,31].

In the light of this, laboratories performing cefiderocol AST by disk diffusion must not only standardize their procedures but also report results with awareness of their diagnostic limitations. Media selection, zone reading technique, and interpretive strategy all contribute to the final susceptibility calls. Based on our results, disk diffusion for cefiderocol remains a feasible method when used carefully, but it should be supported by secondary validation in borderline or ambiguous cases [32,33].

Finally, it is crucial to link these technical observations to clinical microbiology workflows and stewardship strategies. The accurate and reproducible interpretation of AST results for cefiderocol directly affects treatment plans for patients with MDR infections, especially those involving bloodstream or invasive isolates of *K. pneumoniae*. These conclusions align with the resistance phenotypes reported in 2025 by Bedenić et al., where cefiderocol-resistant *K. pneumoniae* strains exhibited extensive β-lactamase gene content and broad resistance profiles, including high MICs even for advanced β-lactam/inhibitor combinations [34]. As such, microbiologists, infectious disease specialists, and pharmacists must collaborate in interpreting results that deviate from standard patterns or fall into ambiguous categories. Kammineni et al. similarly demonstrated high concordance between disk diffusion and broth microdilution for cefiderocol across a large collection of carbapenem-resistant isolates, supporting its use as a frontline method in resource-limited environments [35].

Reassessing both our results and recent literature, it becomes evident that borderline inhibition zones—especially those falling within the ATU range—are not rare but rather a frequent diagnostic challenge in cefiderocol testing. Combined with our observed high level of interpretation errors (very major and major), these findings strongly support the notion that disk diffusion alone is insufficient for reliable susceptibility determination in clinical laboratories. Given the diagnostic ambiguity introduced by media-dependent variability and colony morphology, we recommend that broth microdilution remain the sole reference method for definitive cefiderocol susceptibility interpretation, particularly for MBL-producing *K. pneumoniae* and other multidrug-resistant pathogens. This position aligns with recent publications reporting high misclassification rates in the ATU range and emphasizes the importance of confirmatory testing for guiding antimicrobial therapy decisions [10,24,27,28].

### Strengths and Limitations

This study provides a focused evaluation of factors influencing cefiderocol susceptibility testing in a representative set of carbapenem-resistant, MBL-producing *K. pneumoniae* isolates. By including multiple commercial agar media and assessing inhibition zone interpretation strategies in detail, the study addresses practical variables that can impact the accuracy and reproducibility of disk diffusion results. Notably, it examines the diagnostic ambiguity associated with the presence of isolated bacterial colonies within inhibition zones—an aspect rarely studied systematically despite its potential clinical significance. The role of the ATU is also critically considered, drawing on both experimental data and interpretive challenges encountered during routine testing.

Several limitations should be acknowledged. The number of isolates (*n* = 50), while sufficient for internal comparisons, limits generalizability across broader epidemiological settings. All isolates originated from a single geographic region and clinical context, which may affect applicability to other settings with differing resistance mechanisms. Furthermore, the study did not include genotypic analyses of resistance determinants such as carbapenemase variants, porin mutations, or siderophore receptor profiles, which could influence zone morphology and susceptibility classification. This limitation may be particularly relevant given the heterogeneity of MBL enzymes and the potential contribution of co-resistance mechanisms, porin loss, or altered siderophore uptake pathways. Inclusion of molecular profiling could have elucidated the underlying drivers of phenotypic variability across media conditions. Despite these limitations, the study contributes relevant data to the ongoing refinement of AST protocols for cefiderocol and highlights the need for critical evaluation of interpretive strategies in the presence of methodological uncertainty.

## 4. Materials and Methods

Disk diffusion testing was performed on 50 clinical *Klebsiella pneumoniae* MBL isolates following the EUCAST methodology [23]. MBL status for each isolate was confirmed phenotypically in routine clinical microbiology using EDTA-based synergy tests.

The disks with the cefiderocol (Thermo Fisher Scientific, Waltham, MA, USA) were applied to the surface of the inoculated MHA plates (Thermo Fisher Scientific, Waltham, MA, USA). All isolates were prepared by suspending freshly grown colonies in 0.85% NaCl and adjusting to a turbidity of 0.5 McFarland using a densitometer. The suspensions were used within 15 min of preparation to ensure consistency. All disk diffusion plates were incubated aerobically at 35 ± 1 °C for 18–20 h. The inhibition zone diameters were measured manually with a caliper according to the EUCAST reading guide for the disk diffusion method [23]. Interpretations were based on 2025 EUCAST and CLSI breakpoints [11,12], as summarized in Table 1.

Isolates were previously cultured on 3 different kinds of microbiological media: (1) Columbia Agar with 5% blood, (2) MacConkey Agar, (3) Chromogenic Agar; acquired from 3 manufacturers: Graso Biotech (Starogard Gdański, Poland), Thermo Fisher (Waltham, MA, USA), and Becton Dickinson (Franklin Lakes, NJ, USA)–the formulations for the tested media were presented in Table 4.

Broth microdilution method was used as a reference control, using UMIC Cefiderocol assay (MERLIN Diagnostika, Bornheim, Germany) according to the manufacturer’s instructions. All assays were incubated aerobically at 35 ± 1 °C for 18–20 h.

All bacterial isolates originated from the proprietary collection of the Department of Microbiology and Laboratory Medical Immunology at the Medical University of Lodz, previously obtained from the Medical Microbiology Laboratory of the Central Clinical Hospital of the Medical University of Lodz. All strains, after prior anonymization to prevent their assignment to specific patients, were securely stored in cryobanks at −80 °C. For this research purpose, they were regenerated on the Tryptic Soy Agar (Graso Biotech, Starogard Gdański, Poland), for 18–24 h at 35 ± 1 °C.

### 4.1. Assessing the Isolated Colonies Within the Inhibition Zone

During the visual assessment of inhibition zones in disk diffusion testing, a subset of isolates demonstrated isolated bacterial colonies growing within the otherwise clear inhibition zones different agar media (see raw data in Appendix A). These occurrences were documented and further categorized into three morphological patterns:Complete absence of isolated colonies (Figure 3A);Presence of fewer than 10 isolated colonies within the inhibition zone (Figure 3B);Presence of more than 10 isolated colonies within the inhibition zone (Figure 3C).

Based on this categorization, additional subgroup analyses were conducted to evaluate how different interpretations of colony presence influenced correlation with MIC values and categorical agreement with broth microdilution.

### 4.2. Antimicrobial Susceptibility Testing Errors

AST errors were classified and analyzed following international standards based on comparison to the broth microdilution reference method. For each isolate, disk diffusion results were interpreted using both EUCAST and CLSI breakpoints [11,12]. Three types of categorical discrepancies were defined:Very Major Error (VME)—a resistant isolate falsely categorized as susceptible by disk diffusion;Major Error (ME)—a susceptible isolate falsely categorized as resistant by disk diffusion;Minor Error (mE)—an intermediate isolate falsely categorized as susceptible or resistant, or vice versa (applicable only to CLSI).

Only those isolate–media combinations falling outside the ATU zones were considered for strict error calculation. ATU results were categorized separately due to their intrinsic interpretive ambiguity.

### 4.3. Data Analysis

In this study, statistical analyses were conducted to evaluate differences in inhibition zone diameters and their relationship to MIC values. Given that measurements were obtained from the same bacterial strains across different conditions, paired *t*-tests were applied to assess significant differences between inhibition zone values across various media and manufacturers. To examine the correlation between MIC (broth microdilution) and inhibition zone diameters, Spearman’s rank correlation coefficient was used. This non-parametric test was chosen due to its robustness against non-normal data distributions and its ability to detect monotonic relationships. All statistical tests were two-tailed, and significance was set at *p* < 0.05. All calculations were performed using Python’s SciPy library (version 1.10.1), and visualizations were generated using the matplotlib library (version 3.6.3).

### 4.4. Ethical Issues

The study was conducted in accordance with Good Clinical and Laboratory Practice Guidelines and the Declaration of Helsinki. There was no need to have the consent of the Bioethics Committee to conduct this study, because in the light of the law in force in Poland, the study was not a “medical experiment”–no patient information was included.

## 5. Conclusions

Our findings highlight the diagnostic complexity of cefiderocol susceptibility testing in *K. pneumoniae* MBL. Variability introduced by agar media composition and the interpretation of isolated colonies within inhibition zones affected zone diameter measurements and susceptibility classification. A considerable proportion of results fell into the ATU, underscoring the risk of misclassification when relying solely on disk diffusion. Based on these data, we strongly recommend that broth microdilution remain the reference method for interpreting cefiderocol susceptibility, especially in borderline cases or in the presence of morphological anomalies. Disk diffusion may still be used in resource-limited settings, but only if carefully standardized using validated media, precise zone reading techniques, and consistent interpretive frameworks that account for ATU zones. Clinical microbiologists should interpret ambiguous or borderline results with caution and ideally consult reference methods or expert guidance to ensure accurate patient management.

## Figures and Tables

**Figure 1 antibiotics-14-00527-f001:**
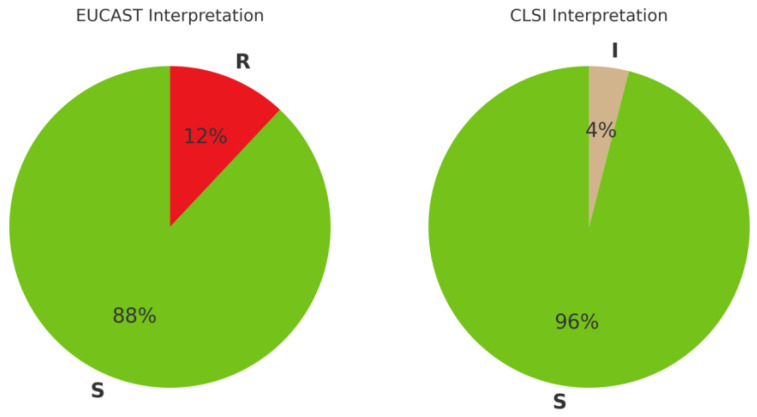
Cefiderocol susceptibility in *Klebsiella pneumoniae* MBL (*n* = 50) using broth microdilution method according to EUCAST and CLSI 2025 standards [10,11] (S—susceptible, R—resistant, I—intermediate).

**Figure 2 antibiotics-14-00527-f002:**
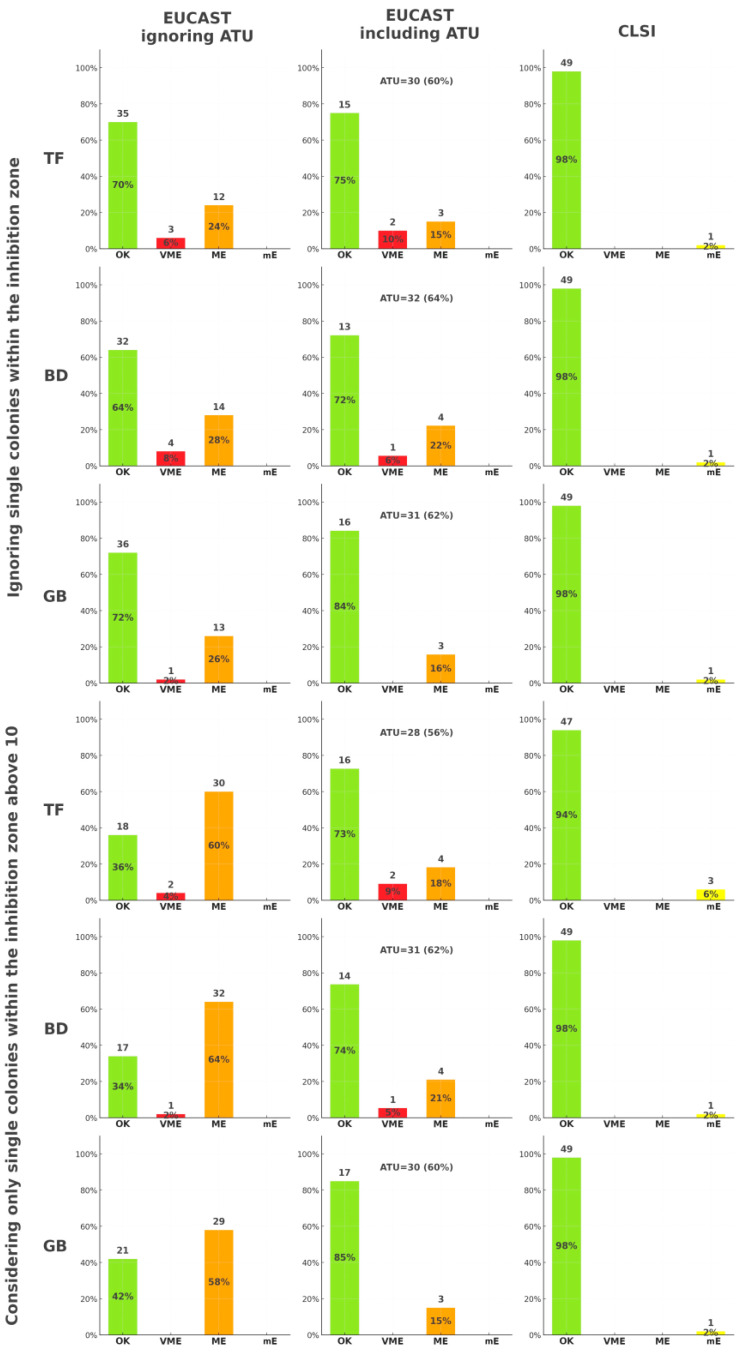
Accuracy of cefiderocol disk diffusion susceptibility testing in *Klebsiella pneumoniae* MBL (*n* = 50) compared to broth microdilution as reference, according to EUCAST and CLSI 2025 standards [10,11] (OK—categorical agreement, VME—very major error, ME—major error, mE—minor error, ATU—area of technical uncertainty; manufacturers: TF—Thermo Fisher, BD—Becton Dickinson, GB—Graso Biotech).

**Figure 3 antibiotics-14-00527-f003:**
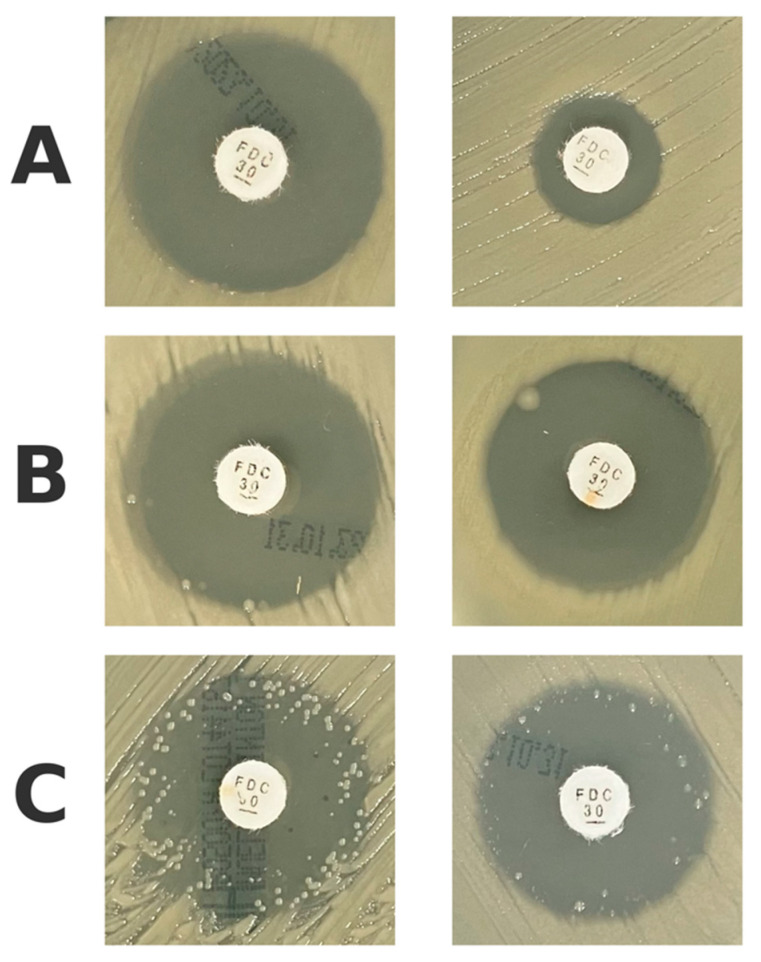
Morphological patterns observed within inhibition zones: (**A**)—complete absence of isolated colonies; (**B**)—presence of fewer than 10 isolated colonies within inhibition zone; (**C**)—presence of more than 10 isolated colonies within inhibition zone.

**Table 1 antibiotics-14-00527-t001:** Current antimicrobial susceptibility testing breakpoints for cefiderocol in Enterobacterales, adapted from EUCAST and CLSI 2025 guidelines [11,12] (MIC—minimal inhibitory concentration; S—susceptible; I—susceptible, increased exposure <EUCAST> or intermediate <CLSI>; R—resistant; ATU—area of technical uncertainty).

Organization	MIC Breakpoints [mg/L]			Zone Diameter Breakpoints [mm] (Disk Content 30 µg)			
	S	I	R	S	I	R	ATU
EUCAST	≤2	-	>2	≥23	-	<23	21–23
CLSI	≤4	8	≥16	≥16	9–15	≤8	-

**Table 2 antibiotics-14-00527-t002:** Comparison of inhibition zone diameters between different agar media and manufacturers under various single colonies within the inhibition zone consideration approaches (media: CAB—Columbia Agar with 5% Blood, MCA—MacConkey Agar, CHA—Chromogenic Agar; manufacturers: TF—Thermo Fisher, BD—Becton Dickinson, GB—Graso Biotech; statistics: t—T-Statistics, *p*—*p*-Value; asterisk means a statistically significant difference).

Agar Medium 1	Agar Medium 2	Ignoring Single Colonies Within the Inhibition Zone		Considering All Single Colonies Within the Inhibition Zone		Considering Only Single Colonies Within the Inhibition Zone Up to 10		Considering Only Single Colonies Within the Inhibition Zone Above 10	
		t	*p*	t	*p*	t	*p*	t	*p*
CAB, TF	CAB, BD	0.286	0.775	−0.007	0.995	0.325	0.746	−0.129	0.897
CAB, TF	CAB, GB	0.332	0.741	−0.390	0.697	−0.080	0.937	−0.064	0.949
CAB, TF	MCA, TF	0.153	0.879	−0.797	0.428	−0.619	0.537	−0.105	0.916
CAB, TF	MCA, BD	0.139	0.889	−0.530	0.597	−0.273	0.786	−0.202	0.840
CAB, TF	MCA, GB	0.061	0.951	−2.185	0.031 *	−1.734	0.086	−0.539	0.591
CAB, TF	CHA, TF	−0.291	0.771	0.462	0.645	0.418	0.677	−0.138	0.890
CAB, TF	CHA, BD	−0.954	0.342	−0.865	0.389	−0.818	0.415	−0.909	0.366
CAB, TF	CHA, GB	−1.089	0.279	−1.432	0.155	−0.971	0.334	−1.553	0.124
CAB, BD	CAB, GB	0.052	0.959	−0.369	0.713	−0.394	0.695	0.070	0.944
CAB, BD	MCA, TF	−0.141	0.888	−0.760	0.449	−0.900	0.370	0.027	0.979
CAB, BD	MCA, BD	−0.147	0.883	−0.507	0.613	−0.557	0.579	−0.083	0.934
CAB, BD	MCA, GB	−0.232	0.817	−2.093	0.039 *	−1.931	0.056	−0.452	0.652
CAB, BD	CHA, TF	−0.586	0.560	0.454	0.651	0.081	0.935	−0.020	0.984
CAB, BD	CHA, BD	−1.252	0.214	−0.828	0.410	−1.076	0.284	−0.859	0.392
CAB, BD	CHA, GB	−1.402	0.164	−1.376	0.172	−1.217	0.226	−1.578	0.118
CAB, GB	MCA, TF	−0.191	0.849	−0.402	0.689	−0.526	0.600	−0.044	0.965
CAB, GB	MCA, BD	−0.196	0.845	−0.166	0.868	−0.195	0.846	−0.150	0.881
CAB, GB	MCA, GB	−0.280	0.780	−1.801	0.075	−1.617	0.109	−0.514	0.608
CAB, GB	CHA, TF	−0.626	0.533	0.842	0.402	0.485	0.629	−0.084	0.933
CAB, GB	CHA, BD	−1.280	0.204	−0.486	0.628	−0.726	0.469	−0.908	0.366
CAB, GB	CHA, GB	−1.428	0.157	−1.061	0.291	−0.878	0.382	−1.614	0.110
MCA, TF	MCA, BD	−0.010	0.992	0.205	0.838	0.292	0.771	−0.109	0.913
MCA, TF	MCA, GB	−0.093	0.926	−1.431	0.155	−1.119	0.266	−0.481	0.632
MCA, TF	CHA, TF	−0.456	0.649	1.242	0.217	1.002	0.319	−0.045	0.964
MCA, TF	CHA, BD	−1.138	0.258	−0.103	0.919	−0.228	0.820	−0.886	0.378
MCA, TF	CHA, GB	−1.287	0.201	−0.692	0.491	−0.389	0.698	−1.609	0.111
MCA, BD	MCA, GB	−0.081	0.935	−1.503	0.136	−1.308	0.194	−0.354	0.724
MCA, BD	CHA, TF	−0.435	0.664	0.952	0.344	0.645	0.521	0.056	0.956
MCA, BD	CHA, BD	−1.101	0.274	−0.291	0.771	−0.490	0.625	−0.764	0.447
MCA, BD	CHA, GB	−1.243	0.217	−0.830	0.409	−0.635	0.527	−1.454	0.149
MCA, GB	CHA, TF	−0.362	0.718	2.577	0.011 *	2.050	0.043 *	0.388	0.699
MCA, GB	CHA, BD	−1.040	0.301	1.265	0.209	0.837	0.404	−0.471	0.639
MCA, GB	CHA, GB	−1.183	0.240	0.664	0.508	0.664	0.508	−1.183	0.240
CHA, TF	CHA, BD	−0.676	0.501	−1.293	0.199	−1.176	0.242	−0.772	0.442
CHA, TF	CHA, GB	−0.801	0.425	−1.838	0.069	−1.318	0.191	−1.410	0.162
CHA, BD	CHA, GB	−0.095	0.924	−0.567	0.572	−0.156	0.876	−0.589	0.558

**Table 3 antibiotics-14-00527-t003:** Correlation between minimal inhibitory concentration (MIC) values and inhibition zone diameters across different agar media and manufacturers under various single colonies within the inhibition zone consideration approaches (media: CAB—Columbia Agar with 5% Blood, MCA—MacConkey Agar, CHA—Chromogenic Agar; manufacturers: TF—Thermo Fisher, BD—Becton Dickinson, GB—Graso Biotech; statistics: ρ—Spearman’s correlation coefficient, *p*—*p*-Value; the six most statistically significant correlations were marked with an asterisk).

Agar Medium	Ignoring Single Colonies Within the Inhibition Zone		Considering All Single Colonies Within the Inhibition Zone		Considering Only Single Colonies Within the Inhibition Zone Up to 10		Considering Only Single Colonies Within the Inhibition Zone Above 10	
	ρ	*p*	ρ	*p*	ρ	*p*	ρ	*p*
CBA, TF	−0.504	0.000190 *	−0.153	0.289484	−0.106	0.462408	−0.493	0.000272 *
CBA, BD	−0.493	0.000272 *	−0.053	0.715205	−0.004	0.977778	−0.500	0.000222 *
CBA, GB	−0.612	0.000002 *	−0.157	0.277542	−0.097	0.503243	−0.623	0.000001 *
MCA, TF	−0.464	0.000683	−0.199	0.165289	−0.118	0.413014	−0.487	0.000339
MCA, BD	−0.398	0.004205	−0.261	0.066671	−0.225	0.115777	−0.404	0.003611
MCA, GB	−0.476	0.000482	−0.236	0.098908	−0.236	0.098908	−0.476	0.000482
CHA, TF	−0.464	0.000691	−0.084	0.562990	0.037	0.799958	−0.471	0.000550
CHA, BD	−0.308	0.029583	−0.074	0.607392	−0.027	0.854732	−0.305	0.031158
CHA, GB	−0.482	0.000396	−0.150	0.300036	−0.150	0.300036	−0.482	0.000396

**Table 4 antibiotics-14-00527-t004:** The formulations for the tested microbiological media as stated in the manufacturers’ documentation.

Manufacturer	Becton Dickinson	Graso Biotech	Thermo Fisher
**Media type**	**Columbia Agar with 5% Sheep Blood**		
**Product name**	Columbia Agar with 5% Sheep Blood	Columbia Agar with 5% Sheep Blood	Columbia Agar with Sheep Blood PLUS
**Catalog number**	254,005	1190PD90	PB5039A
**Formulation**	Pancreatic digest of casein 12.0 g/L, Peptic digest of animal tissue 5.0 g/L, Yeast extract 3.0 g/L, Beef extract 3.0 g/L, Corn starch 1.0 g/L, Sodium chloride 5.0 g/L, Agar 13.5 g/L, Defibrinated sheep blood 5%	Enzymatic casein hydrolysate 5.0 g/L Enzymatic hydrolysate of animal tissues 8.0 g/L Yeast extract 10.0 g/L Agar 14.0 g/L Sodium chloride 5.0 g/L Corn starch 1.0 g/L Defibrinated sheep blood 50.0 mL/L	Special peptone 23.0 g/L Starch 1.0 g/L Sodium chloride 5.0 g/L Agar 10.0 g/L Defibrinated sheep blood 50.0 mL/L
**Media type**	**MacConkey Agar**		
**Product name**	MacConkey II Agar	MacConkey Agar + Crystal Violet	MacConkey Agar No. 3
**Catalog number**	254,025	1020PD90	PO5002A
**Formulation**	Pancreatic digest of gelatin 17.0 g/L Pancreatic digest of casein 1.5 g/L Peptic digest of animal tissue 1.5 g/L Lactose 10.0 g/L Bile salts 1.5 g/L Sodium chloride 5.0 g/L Neutral red 0.03 g/L Crystal violet 0.001 g/L Agar 13.5 g/L	Enzymatic gelatin hydrolysate 17.0 g/L Enzymatic casein hydrolysate 1.5 g/L Enzymatic hydrolysate of animal tissues 1.5 g/L Lactose 10.0 g/L Bile salts 1.5 g/L Sodium chloride 5.0 g/L Neutral red 0.03 g/L Crystal violet 0.001 g/L Agar 13.5 g/L	Peptone 20.0 g/L Lactose 10.0 g/L Bile salts No. 3 1.5 g/L Sodium chloride 5.0 g/L Neutral red 0.03 g/L Crystal violet 0.001 g/L Agar 15.0 g/L
**Media type**	**Chromogenic Agar**		
**Product name**	CHROMagar Orientation	CHROMagar Orientation	Brilliance UTI Clarity
**Catalog number**	254,489	1410PD90	PO5159A
**Formulation**	Chromopeptone 16.1 g/L Chromogen mix 1.3 g/L Agar 15 g/L	Chromogenic mixture 1.0 g/L Peptone and yeast extract 17.0 g/L Agar 15.0 g/L	Peptone 9.0 g/L Chromogenic mix 17.0 g/L Tryptophan 1.0 g/L Agar 10.0 g/L

## Data Availability

The original contributions presented in this study are included in the article’s Appendix A. Further inquiries can be directed to the corresponding author.

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
