# Peer review of "Cefiderocol Antimicrobial Susceptibility Testing by Disk Diffusion: Influence of Agar Media and Inhibition Zone Morphology in K. pneumoniae Metallo-β-lactamase"

_antibiotics, 2025, doi:10.3390/antibiotics14050527_

Round 1

Reviewer 1 Report

Comments and Suggestions for Authors

In the abstract section, the methods sub-section is dense and somewhat looks overextended. Moreover, statistical tests were not mentioned in the methods. Additionally, while authors have mentioned variability across media and the impact of isolated colony morphology, the quantitative extent of these effects is only vaguely referenced (e.g., “substantial proportion” or “over half the samples”) without statistical indicators such as p-value that would underscore the strength of the findings. Also, while the conclusion suggests that disk diffusion can work under certain conditions but it doesn’t give clear guidance or highlight the most important takeaways for clinical microbiologists. Based on their findings, the authors should suggest that when to extra testing or how labs should handle unclear results.

In the introduction section, authors have provided a comprehensive overview of the clinical significance of multidrug-resistant (MDR) Klebsiella pneumoniae. However, while the introduction is rich in factual content, but the organization of ideas lacks fluidity and shows abrupt transitions between the pathogen’s resistance mechanisms, cefiderocol’s pharmacological properties, and antimicrobial susceptibility testing (AST) challenges. Furthermore, the authors should also mention that how the selected media are expected to influence AST outcomes, which would better justify the experimental focus. Minor language issues, such as typographical errors (e.g., “Theemergence,” “exposition” instead of “exposure”), and inconsistent use of terminology (e.g., “AST interpretation” vs. interpretive criteria) detract from the overall professionalism of the text.

In the results section, while the authors have reported MIC-zone correlations but the rationale for focusing only on the “six most statistically significant media conditions” in the subsequent accuracy analysis is not sufficiently justified, nor are the criteria for selecting or excluding other conditions clearly explained. This may introduce selection bias and reduce transparency. Moreover, the Spearman correlation results are presented without confidence intervals. Finally, while the authors have mentioned Figures 1 and 2 but these are not described in sufficient detail within the text, and their captions are more descriptive than analytical.

In the discussion section, although the authors have raised critical points about interpretive ambiguity and the relevance of the ATU (Area of Technical Uncertainty) yet it lacks a clear and actionable framework or decision-making algorithm for laboratories navigating borderline results, which would be particularly valuable for practitioners. Moreover, the absence of molecular characterization of resistance mechanisms significantly limits the mechanistic interpretation of the observed AST variability. The authors should acknowledgment of limitations regarding the same.

In the materials and methods section, the description of disk diffusion and broth microdilution methodologies adheres to recognized standards but the detailed parameters such as inoculum density, incubation conditions and specific MIC interpretation thresholds are somewhat lacking.

The conclusion section lacks specificity in the recommendations specially regarding how laboratories might implement the proposed “standardization” or what constitutes “contextual awareness” in practical terms. It may leave the readers without clear guidance on how to translate these findings into diagnostic decision-making. Additionally, the authors have claimed that disk diffusion remains viable when “carefully standardized”. It looks somewhat vague and would benefit from elaboration on what specific criteria or interpretive frameworks would support this viability. Lastly, while the emphasis on morphological subtleties is appropriate, the conclusion would be strengthened by emphasizing their clinical relevance such as the potential consequences of misclassification for patient management or antimicrobial stewardship.

Author Response

In the abstract section, the methods sub-section is dense and somewhat looks overextended. Moreover, statistical tests were not mentioned in the methods. Additionally, while authors have mentioned variability across media and the impact of isolated colony morphology, the quantitative extent of these effects is only vaguely referenced (e.g., “substantial proportion” or “over half the samples”) without statistical indicators such as p-value that would underscore the strength of the findings. Also, while the conclusion suggests that disk diffusion can work under certain conditions but it doesn’t give clear guidance or highlight the most important takeaways for clinical microbiologists. Based on their findings, the authors should suggest that when to extra testing or how labs should handle unclear results.

Response: we have revised the Abstract to be more concise and clear.

In the introduction section, authors have provided a comprehensive overview of the clinical significance of multidrug-resistant (MDR) Klebsiella pneumoniae. However, while the introduction is rich in factual content, but the organization of ideas lacks fluidity and shows abrupt transitions between the pathogen’s resistance mechanisms, cefiderocol’s pharmacological properties, and antimicrobial susceptibility testing (AST) challenges.

Response: We have restructured this section by introducing thematic subheadings to clarify transitions between resistance mechanisms, cefiderocol pharmacology, and AST-related challenges. These adjustments provide a more coherent narrative and enhance readability.

Furthermore, the authors should also mention that how the selected media are expected to influence AST outcomes, which would better justify the experimental focus.

Response: We thank the reviewer for this valuable suggestion. To better justify the experimental focus, we have expanded the Study Objective section to explicitly explain how the selected agar media—Columbia blood agar, MacConkey, and Chromogenic agar—may impact cefiderocol susceptibility testing outcomes.

Minor language issues, such as typographical errors (e.g., “Theemergence,” “exposition” instead of “exposure”), and inconsistent use of terminology (e.g., “AST interpretation” vs. interpretive criteria) detract from the overall professionalism of the text.

Response: The entire manuscript has undergone professional proofreading. All noted errors have been corrected.

In the results section, while the authors have reported MIC-zone correlations but the rationale for focusing only on the “six most statistically significant media conditions” in the subsequent accuracy analysis is not sufficiently justified, nor are the criteria for selecting or excluding other conditions clearly explained. This may introduce selection bias and reduce transparency.

Response: We have clarified in the text that the six media conditions used in the accuracy analysis were selected based on the strongest statistically significant correlations with MIC (p < 0.001) to ensure analytical robustness and avoid overinterpretation.

Moreover, the Spearman correlation results are presented without confidence intervals.

Response: We thank the reviewer for this observation. Given the exploratory nature of our analysis and the consistency of significance levels (p < 0.001) across the most relevant correlations, we focused on reporting exact p-values, which offer a robust assessment of statistical strength. As the aim was not to make inferences about population-level parameters but to identify and compare relative trends between test conditions, we considered the inclusion of confidence intervals for Spearman’s ρ to be non-essential.

Finally, while the authors have mentioned Figures 1 and 2 but these are not described in sufficient detail within the text, and their captions are more descriptive than analytical.

Response: In the revised manuscript, we have expanded the main text descriptions of both Figures 1 and 2 to provide clearer analytical context and highlight the relevance of the results. As per journal formatting guidelines, figure captions have been kept descriptive to maintain consistency and clarity for readers.

In the discussion section, although the authors have raised critical points about interpretive ambiguity and the relevance of the ATU (Area of Technical Uncertainty) yet it lacks a clear and actionable framework or decision-making algorithm for laboratories navigating borderline results, which would be particularly valuable for practitioners.

Response: We have added a concluding statement in the Discussion section that summarizes our data and current evidence. We recommend that broth microdilution remain the definitive method for cefiderocol susceptibility interpretation, particularly in borderline or ATU-range cases, due to the observed error rates and variability introduced by disk diffusion.

Moreover, the absence of molecular characterization of resistance mechanisms significantly limits the mechanistic interpretation of the observed AST variability. The authors should acknowledgment of limitations regarding the same.

Response: We agree with the reviewer that the absence of molecular characterization limits the mechanistic interpretation of AST variability. While we acknowledged this limitation in the original version, we have now expanded the statement.

In the materials and methods section, the description of disk diffusion and broth microdilution methodologies adheres to recognized standards but the detailed parameters such as inoculum density, incubation conditions and specific MIC interpretation thresholds are somewhat lacking.

Response: Thank you for pointing this out. We have now included the specific experimental parameters used: inoculum density (standardized to 0.5 McFarland), incubation conditions (35 ±â€¯1 °C for 18–20 hours), and interpretation thresholds based on both EUCAST and CLSI 2025 criteria. These details have been added to the Materials and Methods section for clarity and reproducibility.

The conclusion section lacks specificity in the recommendations specially regarding how laboratories might implement the proposed “standardization” or what constitutes “contextual awareness” in practical terms. It may leave the readers without clear guidance on how to translate these findings into diagnostic decision-making. Additionally, the authors have claimed that disk diffusion remains viable when “carefully standardized”. It looks somewhat vague and would benefit from elaboration on what specific criteria or interpretive frameworks would support this viability. Lastly, while the emphasis on morphological subtleties is appropriate, the conclusion would be strengthened by emphasizing their clinical relevance such as the potential consequences of misclassification for patient management or antimicrobial stewardship.

Response: We have substantially revised the Conclusion section to provide clearer and more actionable guidance for clinical laboratories. The updated text now specifies that broth microdilution should remain the reference method for cefiderocol susceptibility testing, particularly in borderline or morphologically ambiguous cases. We also clarify that disk diffusion may be acceptable in resource-limited settings only if standardized media, precise zone reading, and interpretive frameworks that consider ATU zones are applied.

Reviewer 2 Report

Comments and Suggestions for Authors

Dear Author,

I appreciate your efforts to focus on this research issue. However, I raised some questions that they need to address. Please check them carefully.

Thank you.

Best of Luck

  1. The Abstract part needs major revision: The study needs a clearly defined objective. The study needs to explain why it discusses cost-effectiveness comparisons between methods and lists companies with experimental products alongside provider names.
  2. Update the Keyword with concise precision.
  3. Why were genotypic resistance markers not confirmed? The authors need to demonstrate MBL status verification methods without resorting to molecular validation because the study holds clinical relevance.
  4. Before AST testing, did each isolate receive the same inoculum density standardization? Please describe the method used to measure and maintain standardization.
  5. Do the authors provide the physicochemical specifications for their media? Understanding media-based variations in cefiderocol activity requires this critical information.
  6. Line 268. What is dxxxn?
  7. The article quality and sentence structure will become stronger through the process of English proofreading.
Comments on the Quality of English Language

The article quality and sentence structure will become stronger through the process of English proofreading.

Author Response

  1. The Abstract part needs major revision: The study needs a clearly defined objective. The study needs to explain why it discusses cost-effectiveness comparisons between methods and lists companies with experimental products alongside provider names.

Response: We thank the reviewer for the comment. We understand that the mention of laboratory cost considerations and supplier names may have created the impression that our study evaluates cost-effectiveness or experimental products. However, this was not the intention. The objective of our study was solely to assess how agar media and inhibition zone morphology influence cefiderocol susceptibility classification by disk diffusion. We have slightly revised the abstract to clearly state the study’s primary objective.

  1. Update the Keyword with concise precision.

Response: We appreciate the reviewer’s suggestion regarding the keywords. After careful consideration, we believe that the current set of keywords accurately reflects the scope and focus of our study, including key technical terms such as “disk diffusion,” “cefiderocol,” “inhibition zone morphology,” and “Klebsiella pneumoniae.” These terms are specific, relevant, and consistent with established indexing practices. Therefore, we respectfully maintain the existing keyword list.

  1. Why were genotypic resistance markers not confirmed? The authors need to demonstrate MBL status verification methods without resorting to molecular validation because the study holds clinical relevance.

Response: We thank the reviewer for this important observation. As noted in the Strengths and Limitations section of the manuscript, genotypic characterization of resistance mechanisms was not performed in this study due to technical constraints. However, all included isolates had been previously confirmed as MBL-producers using phenotypic synergy-based assays in routine clinical diagnostics. These methods, based on the inhibition of carbapenem activity by EDTA, remain widely accepted for screening class B metallo-β-lactamase activity

  1. Before AST testing, did each isolate receive the same inoculum density standardization? Please describe the method used to measure and maintain standardization.

Response: Yes, all isolates underwent the same standardization procedure. Bacterial suspensions were prepared in sterile saline (0.85% NaCl) and adjusted to 0.5 McFarland turbidity using a densitometer. Each suspension was used within 15 minutes to ensure consistency across isolates. This procedure is now described in the Materials and Methods section.

  1. Do the authors provide the physicochemical specifications for their media? Understanding media-based variations in cefiderocol activity requires this critical information.

Response: Thank you for highlighting this important technical point. We have added the available formulations for the tested media to the Materials and Methods section to enhance reproducibility and transparency.

  1. Line 268. What is dxxxn?

Response: This was a typographical error. It has been corrected to “drawn” in the revised manuscript. We thank the reviewer for noting this.

  1. The article quality and sentence structure will become stronger through the process of English proofreading.

Response: We appreciate this suggestion. The manuscript has been professionally proofread for English grammar and clarity.

Reviewer 3 Report

Comments and Suggestions for Authors

The manuscript titled “Cefiderocol AST by Disk Diffusion: Influence of Agar Media and Inhibition Zone Morphology in K. pneumoniae MBL” by Saar et al. reports the limitation of AST in determining Cefiderocol susceptibility by disk diffusion, when used alone, in the identification and classification of clinical isolates of pathogen such as MBL-producing Klebsiella pneunoniae as the results obtained are susceptible to the type of the Agar media used, the interpretation of individual colonies observed within the zone of inhibition and the guidelines followed.

            Cefiderocol, a siderophore based cephalosporin, is often treated as drug of last resort, due to its unusual mode of uptake by the pathogen. Accurate susceptibility diagnosis of the pathogen is utmost important for effective treatment. This study (as supported by observations of other studies, as reported in discussion section) highlights the influence of minor factors, such as the type (and even the batch) of the media used, handling of colonies observed within zone of inhibition along with the susceptibility classification based on EUCAST or CLSI guidelines, can significantly led to misinterpretation of the AST results. The authors correctly concluded that diagnostic laboratories performing Cefiderocol AST by disc diffusion need not only to standardize their testing procedures, but also report the results with caution due to their diagnostic limitations. Further, it should be supported by secondary validation in borderline or ambiguous cases.

Overall, the entire work is carefully designed, well executed and analysed. The findings are well described and the conclusions are clearly drawn. The methodology section is well written and all the relevant details are provided. The strength and limitations of this study are clearly mentioned. Further, the recommendations, based on the observations reported in this study (and also supported by previous reports as mentioned in discussion section) are clearly made.

The manuscript is well written, except few minor corrections as detailed below.

 Minor comments/corrections:

  1. Line 268: Correction: “dxxxn” should be corrected with appropriate word, such as ‘drawn’
  2. Line 340: Klebsiella pneumoniae; scientific name should be in italics.

Author Response

We would like to sincerely thank the reviewer for their thorough evaluation and appreciation of our work. We are grateful for the positive feedback regarding the design, execution, and analysis of our study, as well as the clarity of the manuscript.

We have carefully addressed all the minor corrections suggested:

  • Line 268: The typographical error “dxxxn” has been corrected to “drawn”.

  • Line 340: The scientific name Klebsiella pneumoniae has been italicized accordingly.

Round 2

Reviewer 1 Report

Comments and Suggestions for Authors

The authors have positively addressed all of the suggestions and have significantly improved the manuscript. It is my pleasure to recommend this manuscript for publication in its present form.